# What Factors Predict the Use of Coercive Food Parenting Practices among Mothers of Young Children? An Examination of Food Literacy, Disordered Eating and Parent Demographics

**DOI:** 10.3390/ijerph181910538

**Published:** 2021-10-08

**Authors:** Lyza Norton, Joy Parkinson, Neil Harris, Laura M. Hart

**Affiliations:** 1Department of Social Marketing, Griffith University, 1 Parklands Drive, Southport, QLD 4215, Australia; j.parkinson@griffith.edu.au; 2Department of Public Health, Griffith University, 1 Parklands Drive, Southport, QLD 4215, Australia; n.harris@griffith.edu.au; 3Centre for Mental Health, Melbourne School of Population and Global Health, University of Melbourne, 207 Bouverie Street, Carlton, VIC 3010, Australia; lhart@unimelb.edu.au

**Keywords:** food parenting practices, food literacy, disordered eating, mothers

## Abstract

Parents have the most significant influence on the development of young children’s eating patterns. Understanding what parental factors best predict specific negative feeding practices is important for designing preventive interventions. We examined the relationship between parents’ use of coercive food parenting practices (pressure to eat and restriction) and parents’ disordered eating, food literacy, Body Mass Index (BMI) and socio-economic status (SES). Adult mothers, with a mean age of 33 years, at least one child aged between 6 months and 5 years and living in Australia (*n* = 819) completed an online questionnaire. Regression models were used to examine predictors of pressure to eat and restriction, respectively. Although the amount of variance accounted for by the models was small, maternal eating disorder symptoms were found to be the most important predictor of coercive food parenting practices. This finding has implications for early nutrition education, which has traditionally focused heavily on nutrition literacy. Parental disordered eating may be a more important preventive target and thus including behavioral strategies for positive feeding practices may better assist mothers in promoting positive eating habits with their children, rather than traditional approaches that aim to increase nutrition literacy.

## 1. Introduction

Disordered eating is a broad term that encapsulates a range of negative or unhealthy eating behaviors and cognitions [1,2], such as restricting intake (i.e., “dieting”). Parents have the most significant influence on the learning and development of children’s positive and negative eating patterns [3]. Negative eating patterns have led to a rise in intergenerational disordered eating in families [4]. Early life and the transition to family foods therefore represent a significant opportunity to establish a foundation of positive eating behaviors, if parents can be supported to nurture positive food parenting practices, such as encouraging children to listen to their body cues around hunger and satiety. However, parents often feel confused and uncertain about “how” to promote eating behaviors to their young children [5,6]. The field of public health nutrition has long focused on increasing parents’ food literacy, as a pathway to improve positive feeding practices in parents and healthy eating patterns in children [7]. This means that there is a plethora of nutrition-related advice for new parents on feeding children. However, parents are reported to find this information difficult to navigate [8] and much of the content is focused on the “what” and “when” to feed young children, with substantial gaps existing for the “how” [7,9].

Examination of the feeding process is via measurement of parental feeding behaviors termed *food parenting practices* (FPPs) [10]. FPPs can influence the health and well-being of the child either positively or negatively [11,12]. One domain of FPPs includes negative *coercive control* practices. These involve parents using pressure and dominance to influence children’s eating behavior. Two of the most studied coercive control practices are *pressure to eat* (i.e., when parents continue to use repeated prompts, despite a child communicating they have eaten enough) and *restriction* (i.e., when parents use parent-centred, authoritarian rules to prevent access to or consumption of certain foods) [13].

Young children are instinctively responsive to internal cues of appetite and satiety [14]. Such responsiveness has been shown to be disrupted by parents using pressuring or restricting behaviors [10]. For example, parents pressuring children to eat, for fear of them not receiving adequate nutrition, can become an entrenched habit. Over time, the child’s opportunities to retain a focus on feelings of fullness and hunger may be minimal, as they are instead shaped to focus on external cues such as finishing food on the plate. This can lead to overeating, emotional or binge eating as children develop and have increasing autonomy over their own food choices [15]. Studies reveal that restrictive parenting practices also predict disordered eating in children [15] including eating in the absence of hunger [13], emotional overeating [16] and an increased intake of the foods parents are restricting when they become available [17].

Food literacy is defined as a person’s capacity to understand, use and interact with food [18]. The concept dominates early feeding education resources for parents [9]. However, a significant gap exists in examining whether food literacy is associated with parental use of coercive control practices. If increasing food literacy is an effective avenue for decreasing coercive control practices, then continued prioritization of food literacy is warranted.

Other parent factors that might be associated with coercive control FPPs include eating disorder symptoms, BMI, and socio-economic status (SES). For example, previous research examining the relationship between parental eating disorder symptoms and restrictive parenting practices [19,20] found positive associations between parental disordered eating and some coercive practices. However, more work is required to examine different populations and a range of coercive practices.

Similarly, some research has found that parental weight status is a predictor of using coercive practices [21]. However, a recent systematic review [22] found no difference between parental BMI status and parents’ use of pressure to eat practices. Given the lack of clarity, it is important to continue to explore both parental eating disorder symptoms and BMI as they relate to coercive practices.

Conflicting results are also present in the literature on parent SES and coercive feeing practices. Inverse relationships between socio-economic status and parental pressure to eat, and to restriction, have been previously reported in one study [23], but no association was found between parental socio-economic status and either coercive practice in another [24]. These two studies, however, were conducted in very different environments—the United States of America and Egypt, respectively—suggesting that further examination is warranted to better understand how parental SES impacts on coercive feeding practices.

This study aimed to examine the relationship between parents’ use of pressure to eat and restriction with parental food literacy, disordered eating, BMI, and SES. Given this is novel research, we were aiming to explore which of these parental variables had the strongest association with negative coercive control practices.

## 2. Materials and Methods

### 2.1. Design

This study gathered cross-sectional data from Australian parents via an online survey. We focused on mothers of young children (aged 0–5 years) with the aim of informing the future development of a preventive intervention.

### 2.2. Procedure

The survey collected information on demographics, food literacy, disordered eating among parents, and the use of restrictive and pressure to eat food parenting practices. A convenience sample was recruited via Facebook. An online flyer was posted to a variety of parent-focused Facebook groups and individuals between 14 January and 15 April 2021. The flyer was titled “Do you have a child aged 6 months to 5 years?” and let parents know we were recruiting for a study on eating habits, body image and parenting practices. It included an electronic link to the survey (hosted via Limesurvey). Interested parents accessed the survey landing page by clicking on the link. Ethics approval was obtained from the Griffith University Human Research Committee (reference number: 2020/969). The survey was conducted anonymously and no identifying information was collected (e.g., names and contact details).

### 2.3. Study Participants

To be eligible for this study, parents needed to: be older than 18 years, live in Australia, have at least one child between the age of six months and 5 years, and possess English language skills sufficient for survey. There were 1188 responses to the survey; however, 369 of those were incomplete and were excluded from analyses. To be included in analyses, participants needed to meet the above inclusion criteria, answer the question “How many children do you have?” and provide complete data for all the three measures (i.e., food parenting practices, disordered eating symptoms and food literacy). Data were removed from 238 participants as they did not answer “How many children do you have?”, 68 as they did not complete all three questionnaire measures, 29 as their postcodes were not from Australia, and 30 as they had children outside the age range. While the survey was open to all parents, only 11 males completed the survey. Hence, the decision was made to remove these 11 surveys to improve the homogeneity of the sample. Data from 819 women were included in the analyses.

### 2.4. Measures

Data on participants’ characteristics, including parental demographics (age, gender, pregnancy status, marital status, household income, educational status, occupational status, ethnicity, and postcode), number of children, child age (calculated from child date of birth), and self-reported height and weight (used to calculate BMI) were collected. Participants also completed the following validated measures of food parenting practices, food literacy and disordered eating.

#### 2.4.1. Food Parenting Practices

The Children’s Feeding Questionnaire (CFQ) is a 31-item validated tool [25] that measures parent attitudes and behaviors regarding child feeding. The following subscales were administered: “perceived responsibility” (3 items), “concerned about child weight” (3 items), “restriction” (7 items) and “pressure to eat” (4 items). A 5-point response scale (disagree to agree) was used for all items, with lower scores indicating lower levels of the attitude or behavior. For the purposes of this study, only data from the restriction subscale (example item: “I have to make sure my child does not eat too many high-fat foods”) and pressure to eat subscale (example item: “My child should always eat all of the food on her/his plate”) were used. The CFQ is widely used [19,26,27], and considered to have adequate validity [25] and acceptable reliability as per Cronbach’s alpha test, α = 0.70 for pressure to eat and α = 0.73 for restriction, from Birch’s original dataset [25].

#### 2.4.2. Food Literacy

A 10-item scale from The Food Literacy Behaviors tool [28] was used to assess food literacy. An example item includes “How often have you done the following in the last month: Plan meals to include all food groups?” It uses a 4-point frequency response scale (1 = never to 4 = always), with item scores summed to obtain a total food literacy score. Higher total scores indicate higher food literacy levels, with scores ranging from 10 to 40. The tool has been validated in Australia from a dataset of 1007 (82% female), from low to medium socio-economic status, with the majority aged 26–35 years. The tool comprises three factors: plan and manage, selection, and preparation, with Cronbach’s alphas of 0.79, 0.76 and 0.81, respectively [28]. While the scale has been described as comprising three domains, we used the scale as a single composite measure due to the apparent overlap in several items within the factor analysis.

#### 2.4.3. Disordered Eating

The Eating Disorder Examination Questionnaire—Short Form (EDE-QS) [29] is 12-item validate tool to assess eating disorder symptoms over the past week. It uses a 4-point frequency response scale, from 0 (0 days) to 3 (6–7 days), to assess eating disorder symptoms over the past week. An example item includes “Have you gone for long periods of time (e.g., 8 or more waking hours) without eating anything at all in order to influence your weight or shape?” Scores across all 12 items are summed (ranging from 0 to 36), with higher scores representing more eating disorder symptoms. A threshold score of 15 has been suggested as indicating probable eating disorder diagnostic status [30]. The reliability and validity have been established using data (*n* = 559) from an online survey (university students and those identifying as having a history of eating disorders). High internal consistency was found, α = 0.913. The measure is highly correlated with the original 28-item Eating Disorder Examination Questionnaire (EDE-Q), which is considered the gold standard in eating disorder research (r = 0.91 for people without an eating disorder; r = 0.82 for people with an eating disorder). A recent study also revealed excellent reliability, with a Cronbach’s alpha of α = 0.91 [30].

#### 2.4.4. Body Mass Index (BMI)

BMI classifications were in keeping with the World Health Organization (WHO) classification system, (less than 18.5 kg/m^2^, 18.5 to 24.9 kg/m^2^, 25 to 29.9 kg/m^2^ and 30 kg/m^2^ and above) [31]. Mothers’ self-report of height and weight were used to calculate BMI. The majority of the sample (67%) reported their weight as “known”, with the remaining reporting it as an “estimate”. Self-report is a reliable estimate of BMI when compared to anthropometric measures [32].

#### 2.4.5. Socio-Economic Status (SES)

Participants were asked to report their total gross household income (before tax) for the past 12 months. Seven response options were provided, in line with recommendations from the Australian Bureau of Statistics [33] (Less than $25,000, $25,000–$50,000, $50,001–$75,000, $75,001–$100,000, More than $100,000, I am not sure, and Prefer not to say). Categories were collapsed into the four analyses (low income ≤$50,000, middle income $50,001 to $100,000, upper income >$100,000, and Other “Not sure/prefer not to say”).

### 2.5. Data Analysis

Initial cleaning of data was undertaken using Open Refine, which included removing incomplete data and those that did not meet inclusion criteria. The data were screened for assumption testing and analyzed in SPSS version 27. The following assumptions were met prior to the analysis being performed: the dependent variables were continuous, normal distribution was apparent in the dependent variables, the predictor variables were continuous or in the case of income variables (categorical) converted to dummy variables and linear relationships existed between the dependent and predictor variables. No multicollinearity existed between the predictor variables, as per the variance inflation factors (<1.30) and the condition index ranged between 1.00 and 21.82. The Breusch–Pagan test for both dependent variables (pressure to eat and restriction) with the predictor variables was used and both *p* values were >0.05 (0.07 and 0.811, respectively), indicating that the assumption for homoscedasticity was met. A series of multiple linear regression analyses were performed between the two dependent variables (pressure to eat and restriction) and predictor variables (food literacy, disordered eating, BMI and SES). Such analysis was based on simultaneous entry of the predictor variables, where all the predictor variables were entered into the equation at the same time. This was an appropriate method as there was a small set of predictors, all of which had limited or no past analysis.

## 3. Results

Table 1 outlines participant demographic characteristics. On average, women were aged 33 years (SD: 6.50). Most of the participants were married (73%) and had a bachelor’s degree or higher (70.6%). Just over half (51.8%) were in full-time or part-time employment. A large proportion (76.2%) of the sample identified as Australian, with just over half (56.7%) having two children (See Table 1).

On average, mothers’ total pressure to eat score was 4.42 (SD: 0.64). Mothers’ average restriction score was 3.08 (SD: 0.91) of a possible 5.

Table 2 outlines the food literacy mean score was 27.95 (SD: 4.46) of a possible 40, indicating moderate literacy scores. The mean score for the EDE-QS was 8.08 (SD: 6.61) of a possible 36, suggesting mild to moderate subclinical, eating disorder symptoms. Table 1 also shows the proportion of participants in each BMI and SES category. 42.7% of the sample reported a BMI range of between 18.6 and 24.9, which is a larger proportion than the Australian national figure of 31.7% [34]. The majority of participants were of high SES, with 58% reporting their annual income was $100,000 or above.

Table 3 shows the outcomes for pressure to eat and all predictor variables. There was a significant negative relationship between pressure to eat and food literacy. There was a significant positive relationship between pressure to eat and eating disorder symptoms. A significant negative relationship existed between pressure to eat and BMI. Both the low-income category (<$50,000) and Other (“not sure/not say”) category had a significant positive relationship with pressure to eat, compared to the reference category of >$100,000. The amount of variance in pressure to eat scores explained by the four variables was 7.9% (ANOVA F = 7.666, *p* < 0.001).

Table 4 shows the outcomes for restriction and all predictor variables. Food literacy and restriction showed no statistically significant relationship. There was a significant positive relationship with eating disorder symptoms and a significant negative relationship with BMI. No income category showed statistically significant relationships with restriction. The amount of variance in restriction scores explained by the four variables was 3.8% (ANOVA F = 3.510, *p* < 0.001). 

## 4. Discussion

This study aimed to examine which parental factors are the strongest predictors of parents’ use of pressure to eat and restriction. Our results highlight that maternal eating disorder symptoms were more important predictors of coercive food parenting practices (pressure to eat and restriction) than food literacy, BMI, or SES. The overall predictive value of the variables combined, however, was low for both coercive behaviors, indicating that they are only part of the complex feeding dynamic between parents and children.

Our findings revealed a significant negative relationship between food literacy and pressure to eat. However, we found no significant relationship between food literacy and restriction. This lack of association highlights that other parental factors may be more important to the interplay of coercive practices than food literacy. Perhaps restriction behaviors stem from parents’ own restrictive or “dieting” tendencies, rather than parental food literacy skills. Such restrictive behaviors may be inadvertently transferred to parental food parenting practices. Cross-sectional research supports this theory, finding positive associations between maternal dieting habits and children’s dieting practices [35,36]. Longitudinal research corroborates these findings, revealing maternal dieting to be a significant predictor of child drive for thinness, in a 20 year follow up study [37], hence, the need to understand more about maternal disordered eating behaviors and food parenting practices. This is an important and novel finding as the literature is sparse on the relationship between food literacy and coercive feeding, yet our results point to a need to review the importance placed on developing food literacy as a preventive intervention. The use of food literacy concepts in early feeding education for parents is widespread [9]. Being food literate is potentially a beneficial starting point for parents to develop skills and knowledge to feed their children. However, a review of parental feeding-related interventions [38] argued that feeding-related education is often provided too late and is not comprehensive, imploring researchers to create quality early feeding advice. Our results show that mothers with higher food literacy have reduced likelihood of using pressure to eat feeding practices, but not of using restriction, which is also associated with important negative outcomes for children.

Our findings indicated that the higher a mother’s eating disorder symptoms, the greater her coercive practices (pressure to eat and restriction) were. Consistent with our results, other studies (also examining younger children) have found an association between maternal eating pathology and food parenting practices [19,20,39]. Haycraft and Blissett (2008) also used the subscales of pressure to eat and restriction from The Children’s Feeding Questionnaire [25] and measured maternal drive for thinness, bulimia and body dissatisfaction [19]. Findings revealed mothers’ bulimia scores were positively correlated with restriction of their daughter’s intake (mean age 3.5 years). Restriction was also positively correlated with uncontrolled eating in a study by Musher-Eizenman and colleagues (2009), with a similar child age group (mean age 5 years) [20]. Consistent with our study, the research design was cross-sectional, suggesting a positive relationship between mothers’ eating behaviors and use of coercive practices.

Parental coercive behaviors provide children with less opportunity to develop self-regulatory skills [40]. Extending on previous studies, our research supports the evidence that mothers’ disordered eating behaviors are specifically predictive of both pressure to eat and restriction. Across both subscales, we found eating disorder symptoms to be the most significant predictor compared to food literacy, BMI, and SES. Assisting parents in preventing children from developing disordered eating and negative relationships with food requires a clear understanding about what parental factors are likely to be predictive of both positive and negative behaviors. This finding suggests broadening our education and support for parents of young children to include guidance on feeding behaviors that promote positive eating habits, especially in the context of parent disordered eating.

Our results show that parental BMI had a significant negative relationship with pressure to eat and restriction; as BMI scores increased, pressure to eat and restriction practices decreased, demonstrating that higher weight parents in our sample used less coercive feeding practices. Prior research on food parenting practices and parental BMI shows mixed results. Several cross-sectional studies have found no significant relationship between food parenting practices (as measured by CFQ) and parental BMI [41,42,43]. Given the small amount of variance accounted for by BMI, it is possible that smaller sample sizes than our 819 would be under-powered to detect this effect. In contrast though, Gray and colleagues found restrictive feeding practices to be positively correlated with parental BMI, and this relationship was moderated by parent body dissatisfaction [44], which is a known risk factor for disordered eating symptoms, such as skipping meals and binge eating. The results of Gray and colleagues may be in keeping with our own findings that parental symptoms of disordered eating are predictive of restrictive feeding practices, and more strongly so than parental BMI. We believe that our findings underscore the need to move on from using BMI as a predictor of nutritional or health status, given other factors (such as disordered eating) are likely more valid and useful. Eating behaviors are complex and interactive, hence the relationships that predict parental feeding practices are also likely to be more dynamic than BMI. We used BMI in this study as a measure of variability among parents that may impact on the use of FPPs, without hypothesizing a mechanism of action. BMI is, however, often used as a proxy for nutritional or health status, even though they are poorly correlated and public health experts suggest ceasing the use of BMI in this way [45]. Future research could consider using a Food Frequency Questionnaire [46] to provide insights on how parental dietary intake impacts on FPPs, especially in the context of where parents are exhibiting disordered eating, to understand whether it is the context or the content of feeding practices that most impacts on children’s eating outcomes.

Our study showed a significant positive relationship with both the lowest-income (<$50,000) and “not sure/Not say” categories compared to the reference category of >$100,000 and use of pressure to eat. In other words, among those families with less financial resources, and those not able or willing to quantify their financial resources, parents were more likely to pressure their children to eat more. Very limited research exists on the direct relationship between parental SES and food parenting practices. In one study, higher consumption of soft drinks in children from lower SES, compared to higher SES families, was found [47]. However, this result was almost entirely mediated by specific parenting practices (e.g., accessibility and permissiveness), suggesting that the behaviors exhibited by the parents were the crucial predictor of children’s soft drink intake, as opposed to their SES.

There are several limitations to this study. As this study is cross-sectional in design, causal relationships cannot to be drawn. Caution is needed when considering the generalizability of the findings as the participants were predominantly from more affluent households and were well educated. As child data were not collected, we were unable to ascertain whether parents’ restriction or pressure to eat practices were more likely according to particular child factors, such as fussy eating, food responsiveness, temperament or gender. Additionally, the possibility of selection bias exists because survey respondents were recruited from internet-based groups and participation was voluntary. Respondent’s likelihood for participating in a study is correlated with interest in the topic of the survey [48]. Therefore, we acknowledge the data may not represent the entire target population, it may be skewed to those with an interest in the topic.

Despite these limitations, this study does increase our understanding of parental factors associated with the use of coercive food parenting practices. In particular, the lack of association between parental food literacy and pressure to eat is a novel finding, considering early feeding education materials for parents are very heavily weighted towards food literacy concepts [7,9]. To assist in the prevention of eating problems in the future, perhaps altering the balance of education content provided to parents is warranted. Strategies gaining evidence include cultivating family connection during mealtimes [49], modelling eating based on hunger and satiety (intuitive eating) [50], using non-stigmatizing language around food and bodies [51], promoting a functional view of the body [52] and responsive feeding [53]. These strategies, if shared with parents of young children, could assist in helping them create environments for their children that enhance positive relationships with food and eating.

For some parents with their own disordered eating symptoms, the line may be blurry between promoting healthy eating choices and unintentionally using coercive, and especially restrictive, practices. Hence, we would suggest that there is value in using a more inclusive concept such as food well-being. Block and colleagues suggested the term of “food well-being” in 2011, defining it as having a positive relationship with food from a psychological, physical, emotional, and social point of view [54]. Such a term would be useful if used in conjunction with to the current concept of food literacy. The definition of food literacy is yet to have agreement, with 51 different definitions cited in a recent systematic scoping review [55]. Additionally, it is vital that well-validated and practical tools are developed to measure food literacy as a more holistic concept. In an age where health promotion is very much targeting prevention of excessive weight, we need to acknowledge that only focusing on the traditional “food literacy” component of education may have unintended consequences, such as promoting disordered eating.

## 5. Conclusions

Our study adds to the small body of research examining which parental factors predict coercive feeding practices. In particular, it highlights that mothers’ disordered eating symptoms are a greater predictor than food literacy, BMI and socio-economic status. Longitudinal research is required to gain a deeper understanding pertaining to causality. By building our understanding of what parental and child factors best predict certain food parenting practices, interventions can be tailored to target specific behaviors. Currently, education resources, for the early years, are mostly developed through the lens of food literacy, with the “what and when” of nutrition education remaining the central messages. More comprehensive resources are required that include behavioral strategies for parents around how to avoid coercive feeding practices. Including high-quality information on behavioral strategies for parents within traditional nutrition education for the early years may lead to more promising outcomes in promoting healthful eating and preventing disordered eating into the future.

## Figures and Tables

**Table 1 ijerph-18-10538-t001:** Demographics of Participants. (*n* = 819).

Variable	*n*	%
BMI		
Less than 18.5	22	2.4
18.5–24.9	350	42.7
25.0–29.9	234	28.6
30.0 and above	200	24.4
Missing	13	1.6
Reported weight		
Weight known	553	67.5
Weight estimate	266	32.5
Pregnant		
Yes	95	11.6
No	724	88.4
Marital status		
Single (never married)	31	3.8
Domestic partnership	150	18.3
Married	598	73.0
Divorced, separated, widow, other	31	3.8
Missing	9	1.1
Household income (gross annual)		
≤$50,000	78	9.5
$50,001 to $100,000	234	26.5
>$100,000	477	58.2
Not sure/not say/other	60	7.3
Educational status		
High school not completed	17	2.1
High school completed	60	7.3
Tafe or trade qualification	75	9.2
Diploma	89	10.9
Bachelor’s degree or above	−578	70.6


Occupational status		
Full time	114	13.9
Part time or casual	334	40.8
Student	135	2.3
On leave (e.g., maternity)	11	16.5
Looking for work	85	1.3
Home duties	85	10.4
Any combination of above	116	14.2
Missing	5	0.6
Ethnicity		
Australian	624	76.2
Mixed	34	4.2
New Zealander	27	3.3
British	23	2.8
Other	111	13.5
Number of children		
1	242	29.5
2	464	56.7
3	113	13.8

**Table 2 ijerph-18-10538-t002:** Means and Standard Deviations on the Measures of Eating Disorder Examination, Food Literacy, Pressure to Eat and Restriction in Mothers (*n* = 819).

Variable	Mean (SD)
Total EDE-QS score	8.08 (6.61)
Total food literacy score	27.95 (4.46)
Pressure to eat	4.42 (0.64)
Restriction	3.08 (0.91)

Total EDE-QS: The Eating Disorder Examination Questionnaire—Short Form (EDE-QS) [29]; Total food literacy: The Food Literacy Behaviors Tool [28]; Pressure to eat: subscale from The Children’s Feeding Questionnaire (CFQ) is a 31-item validated tool [25]; Restriction: subscale from The Children’s Feeding Questionnaire (CFQ) is a 31-item validated tool [25].

**Table 3 ijerph-18-10538-t003:** Multiple regression analysis scoring for food literacy, eating disorder symptoms, BMI and household income predicting level of parental pressure to eat practices.

Variables	B (95% Confidence Interval)	*p*
Total food literacy	−0.030 (−0.046–0.014)	<0.001 **
Total EDE-QS score	0.034 (0.022–0.046)	<0.001 **
BMI	−0.020 (0.032–0.008)	0.001 *
Household Income (gross annual)		
≤$50,000	0.273 (0.023–0.523)	0.032 *
$50,001–$100,000	0.110 (−0.055–0.275)	0.192
Not sure/not say/other	0.278 (0.004–0.552)	0.047 *

R^2^ = 0.077 (7.7%), ANOVA F= 11.089, *p* < 0.001, reference group for income >$100,000; * *p* < 0.05, ** *p* < 0.001; BMI: Body Mass Index; Total EDE-QS: The Eating Disorder Examination Questionnaire—Short Form (EDE-QS) [29].

**Table 4 ijerph-18-10538-t004:** Multiple regression analysis scoring for food literacy, eating disorder symptoms, BMI and household income predicting level of parental restriction practices.

Variables	B (95% Confidence Interval)	*p*
Total food literacy	−0.003 (−0.018–0.011)	0.661
Total EDE-QS score	0.024 (0.022–0.046)	<0.001 **
BMI	−0.015 (0.013–0.035)	0.005 *
Household Income (gross annual)		
≤$50,000	0.273 (0.023–0.523)	0.202
$50,001–$100,000	0.013 (−0.136–0.163)	0.859
Not sure/not say/other	−0.122 (−0.370–0.125)	0.333

R^2^ = 0.031 (3.1%), ANOVA F= 4.261, *p* < 0.001, reference group for income >$100,000; * *p* < 0.05, ** *p* < 0.001; BMI: Body Mass Index; Total EDE-QS: The Eating Disorder Examination Questionnaire—Short Form (EDE-QS) [29].

## Data Availability

The dataset used is available from the corresponding author on reasonable request.

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
