# Peer review of "What Factors Predict the Use of Coercive Food Parenting Practices among Mothers of Young Children? An Examination of Food Literacy, Disordered Eating and Parent Demographics"

_ijerph, 2021, doi:10.3390/ijerph181910538_

Round 1

Reviewer 1 Report

I thank the editor for the opportunity of reviewing this manuscript. The topic of "food parenting practices" is very important for early prevention eating disorders in the child and in this future adolescent/adult. The paper shows that it is important deep discussion by the health professionals on this topic and probably we must change the way we think and do nutritional education. In general paper is well written and clear, especially the discussion. My suggestions are related to methods and many of them in the results section. I detail below: 1. Introduction - Lines 75-77:"However, if other parent variables are more strongly associated with ...".It is better to present this kind of reflection in discussion and conclusions. Here the authors should present the state of art and the lack if knowledge, at maximum in the final present the hypothesis, but not here. I suggest rewrite the sentence or remove it. - Lines 88-95: I little confusing this part. I suggest to divide the paragraph or let the ideas clearer and do a better transition in the discussed variable.First the BMI and then the SES. - Lines 96-98: It is redundant this information before the objective, it was already mentioned. I suggest the authors present their study hypothesis here. 2. Materials and Methods - Lines109-110: I suggest to put this part of ethics issues in the of this subsection. How did the authors keep confidentiality and anonymity being and online survey? - It is confuse to mention the selection criteria here, better place them in the "population/sample section". another thing related to this topic, this study was for parents or for mothers? Because the fathers/males were excluded and I did not understand why... Maybe it is better to mention that was just for mothers since the beginning. - Line 163: (Begley 2018). Please correct the citation format. - 2.44 Body Mass Index (BMI) - in this section the authors must mention the cut-off points adopted. In table 1 it is presented one, but does not agree to WHO in relation to underweight category. The correct cut-off point is under18.5 kg/m2.Those with BMI18.5 are "normal". Please , corrected it in table 1. - 2.5 Data Analysis: It is missing the details about the regression models. Backward or forward method?How about the multiple linear regression assumptions? How were they tested? It is not sufficient to mention "The data were screened for assumption"(lines 200-01). - Line 203: duplicate BMI. Eliminate one. 3. Results The results must be re-structured. The division titles are not adequate for a results section (3.1-3.4), it is more like methods section. Otherwise, the authors remain in an entire text this section but in a logical order of presentation. - Lines 207-08: Remove this information because you have already presented before. "Of the survey respondents, 819 mothers with at least one child aged between 6 months and 5 years of age, were eligible for inclusion in analyses." - Table 1: It it too big and confusing. I suggest to divide it in 2. One with the participants characteristics and a second one with the eating disorder symptoms. If everybody is female it is not necessary to put it in table. Remove it. Correct BMI classification, as mention before. some variables present so many categories, is it possible to join some of them? In Occupational status, when you say "Combination of above", it is not clear if it is two or more posibilites or which are they. In Ethnicity, what is mixed? - Lines 281-283 and 290-91: The authors present discussion here. it is not the place to put it. Remove it from here and place in the corresponding section. For example: "This is higher than previous research, which found mean pressure to eat scores of 3.51 (SD: 0.75) among mothers with children aged 6 to 8 years in the Netherlands [27]". - Lines 295-7: Remove this information because you have already presented before. This method section. - Tables 2 and 3: all the abbreviations/initials must be in footnote. Please, standardize if it is p or P. The titles are incomplete, rewrite them. Finally, the authors should standardize if they are going to mention as predictors variables or independent variables. The way is presented sometimes seems that are different things, but are not. 4. Discussion: In limitations it is necessary to mention the possible selection bias (internet, just in specific groups of mothers, etc). How did the authors guarantee the veracity of the information ? Is there any possibility of bias? Discuss it.

Reviewer 2 Report

The article „What factors predict the use of coercive food parenting practices among mothers of young children?: An examination of food literacy, disordered eating and parent demographics” presents important findings that may help in planning nutrition education for parents of young children. This is extremely important as it can go a long way in minimizing eating disorders among children. The study was properly planned and executed, and the findings were comprehensively presented, discussed, and debated. From the merits point of view, the work does not raise any objections.

A weakness of the research is that respondents self-reported height and weight (without a trained person taking the measurements). Such a result is unreliable because participants may have under- or overestimated their own body size. Of course, it is understandable that in the age of pandemics there was no other option.

There are also a few editorial comments:

  • in my opinion, periods should be added to the numbering of subsections
  • the text is not justified in some places
  • in the article there is a sliding text that overlaps with the tables

Round 2
